**Data Availability Statement:** Data was collected through a virtual survey on the INEI website from May to August of 2017. The survey data is publicly

# Discrimination and mental health in the minority sexual population: Cross-sectional analysis of the first peruvian virtual survey

**David R. Soriano-Moreno**[1], **David Saldaña-Cabanillas**[1], **Luigy Vasquez-Yeng**[1], **Javier Antonio Valencia-Huamani**[2], **Jorge Luis Alave-Rosas**[1,3], **Anderson N. Soriano**[1]*

1 Clinical and Epidemiological Research Unit, School of Medicine, Universidad Peruana Unión, Lima, Peru, 2 Asociación Civil Impacta Salud y Educación, Lima, Peru, 3 Department of Internal Medicine, Good Hope Clinic, Lima, Peru

* andersonsoriano@upeu.edu.pe, andsor19@gmail.com

## Abstract

### Objective

This study sought to evaluate the association between discrimination and having mental health problems in the past 12 months in the sexual minority population in Peru.

### Methods

We conducted a cross-sectional analysis of a secondary database corresponding to the first LGBTI survey in Peru in 2017. We included adults who self-identified their sexual orientation as gay, lesbian, bisexual, pansexual, or asexual/others. Both the exposure and dependent variables were self-reported by the participants. Multivariable Poisson regression was used to determine the association by calculating adjusted prevalence ratios (APR) with 95% confidence intervals (95% CI).

### Results

Out of 9760 respondents, more than two-thirds of the participants reported having been discriminated against or having experienced violence at some time in their lives (70.3%) and one-fourth reported having mental health problems (23.8%). In the multivariable regression model, the prevalence of mental health problems in the last 12 months was 72% higher for the group of individuals who experienced discrimination when compared with the group that did not experience discrimination (APR = 1.72, 95% CI 1.57–1.88). The association was stronger among who self-identified lesbians (APR = 2.08, 95% CI 1.65–2.64).

### Conclusion

The prevalence of mental health problems and discrimination was high in this population. In addition, we found a statistically significant association between discrimination and the occurrence of mental health problems in the last 12 months.

available on the INEI website (http://iinei.inei.gob.pe/microdatos/).

**Funding:** ANS is supported by a Fogarty International Center of the US National Institutes of Health Training Fellowship (5D43 TW011502). The funders had no role in study design, data collection and analysis, decision to publish, or preparation of the manuscript.

**Competing interests:** The authors have declared that no competing interests exist.

## Introduction

The LGTBI community (lesbian, gay, bisexual, trans, and intersex) represents people with diverse sexual orientation or gender identities. It constitutes a vulnerable group because they do not fit within society's conventional ideas regarding sexual orientation, which leads them to be ridiculed, intimidated, and even physically abused. Therefore, currently, the United Nations Organization works to protect the LGBTI community against violence and discrimination and declares that all people have the same freedom and equal rights [1]. Similarly, the United Nations Development Program prohibits discrimination based on sexual orientation and gender identity and discrimination based on LGBTI status [2,3]. In Peru, since January 2017 Legislative Decree 1323 entered into force, including as categories protected against discrimination, gender identity, and sexual orientation, being also considered as aggravating in crimes against LGBTI persons [4].

For several decades, the sexual orientation conception of the LGTBI community has tended to be discriminated against. For that reason, LGBTI activism gained force in the 70s which led to the fact that in 1973 the Board of Directors of the American Psychiatric Association voted to remove the diagnosis of "homosexuality" from the DSM III classification of psychiatric disorders. This event was joined by the American Psychological Association, which since 1975 has been working to eliminate the concept of mental illness that has been associated with the sexual orientation of these individuals [5,6]. Finally, in 1990, the WHO removed homosexuality from the ICD-10. In recent years, the historical antecedents and LGTBI activism have managed to change the opinion of society, which has increasingly opposed discrimination. Nevertheless, hostile expressions are still used towards members of the LGTBI community [7]. In Peru, a country with conservative traits, unions between people of the same sex have not yet been officially recognized. Most of the population rejects the implementation of equal marriage. With a government that prohibits discrimination and incitement to discrimination based on sexual orientation and gender identity, but that still maintains Article 333 subsection 9 of the current Civil Code, where homosexuality is considered as a ground for divorce [8]. It fails to curb the high rates of discrimination suffered by the LGBTI community.

In the LGTBI population of the United States, discrimination was associated with mental health problems such as depression and anxiety which were reported more among people self-identified as gay than bisexual or heterosexual individuals. Within women, the self-identified bisexual population was more likely than lesbians or heterosexuals to report disorders and mental health conditions such as depression and/or anxiety. In addition, within men, the self-identified gay population was more likely than the bisexual population to be discriminated against because of their sexual orientation 50% vs 24.2% ($\chi 2 = 11.3$, p < .01), and within women the self-identified lesbian population was more prone than the bisexual or heterosexual population to be discriminated against because of their sexual orientation [9]. Previous studies in the United States have shown that people who are part of this population have a higher prevalence of suicide, depression, anxiety, and substance use disorders [10,11]. Meyer's conceptual framework addressed the association between these variables and hypothesized that factors such as social support, expectations of rejection, concealment, and internalized homophobia could explain the causal pathway [12].

In Peru, there are no studies that evaluate the predisposition to suffer mental disorders in the LGTBI population. However, we observe that the Peruvian trans population is seriously affected. Violence comes, in some regions, from the general population or the Police and other authorities. This is a very difficult situation because this population may also suffer discrimination when reporting [13].

Hence, we conducted a cross-sectional analysis of a secondary database corresponding to the first LGBTI survey in Peru in 2017 to evaluate the association between discrimination and mental health problems in the sexual minority Peruvian population.

## Methods

### Setting and population

A cross-sectional analysis of a secondary database from the First Virtual Survey for LGTBI people in Peru was conducted. This survey was carried out by the Peruvian National Institute of Statistics and Informatics (INEI) in 2017. The main aim of this survey was obtained peruvian LGBTI population statisical information who were over 18 years of age. It had a exploratory design and used non-probabilistic sampling due to no prior information being available on the size of this population to select a sample. Data was collected through a virtual survey on the INEI website from May to August of 2017. The clean survey data is publicly available on the INEI website (http://iinei.inei.gob.pe/microdatos/). The National Directorate of Census and Surveys of INEI was in charge of data collection. The web application was designed for the purpose and subsequently validated by a process involving mixed methods. An analysis of the consistency of the information was also carried out before the creation of the final database. Further details of the data collection process can be found on their web page [14]. Target population was adults between 18–60 years old who self-identified their sexual orientation as gay, lesbian, bisexual, pansexual, or asexual/others. The question that assessed sexual orientation did not have transgender and intersex as answers, as both terms refer to sexual identity.

### Variables

Past-year mental health disorders were assessed using the following question (P105_3) "In the last 12 months, have you had a mental health problem such as (depression, anxiety)?". The answer was dichotomous (yes, no). This question involves any degree of symptomatology from a belief of anxiety or depression to a clinical diagnosis. Discrimination was assessed using one item (P201): "Have you ever suffered discrimination and/or violence?". The answer also was dichotomous (yes, no). Given that there was no question used to measure the type, the frequency, and extent of the discrimination, the understanding is that respondents would report any life experience.

Potential confounding variables were described and included in the multivariable analysis. Sociodemographic variables included education level (university-level studies or not), physiological sex at birth (male, female), race (mestizo, black, white, Indian native), disability (yes, no), and sex work at least once (yes, no) [15–23]. A history of chronic and infectious diseases in the last 12 months was also taken into account with questions "In the last 12 months, have you had any chronic diseases (asthma, chronic bronchitis or emphysema, hypertension, i.e., high blood pressure, diabetes, high blood sugar)" (yes, no) and "In the last 12 months, have you had any infectious diseases (tuberculosis, sexually transmitted infections (syphilis, gonorrhea, etc.), HIV/AIDS)" (yes, no). A question about health insurance had the categories none, SIS = Peruvian Comprehensive Health Insurance and other health insurances.

### Statistical analysis

Absolute and relative frequencies were used to describe categorical data. Median and interquartile range was used to describe the age. Prevalence of discrimination and mental health problems were calculated in the total sample and by sexual orientation. The relationship between the key variables and sexual orientation was assessed using the Chi-squared test. We

calculated unadjusted and adjusted prevalence ratios (PR) with 95% confidence intervals (95% CI) using Poisson regression with robust variance to examine the association between discrimination and mental health problems. The potential confounding variables described above were included in the adjusted model. In addition, we examined the effect of discrimination and sexual orientation simultaneously on mental health. First, we added the interaction between discrimination and sexual orientation to the adjusted model. Then, we calculated the interaction terms multiplying the prevalence ratios as follows: $RP_{Discrimination,Sexual\ identity} = RP_{Discrimination} * RP_{Sexual\ identity} * RP_{Discrimination,Sexual\ identity}$. We choose Poisson regression because the frequency of the outcome event, mental health problems, was greater than 10% and the ORs might overestimate the magnitude of the association [24,25]. A p-value $<0.05$ was considered significant. All analysis was performed using R, version 3.2.5.

## Ethics approval

This study was approved by the Faculty of Health Sciences of the Universidad Peruana Unión (Certificate of Approval: N˚00116-2020/UPeU/FCS/CIISA). This was a secondary analysis of a publicly available database. The database does not contain personal data that would allow for the identification of the respondents, and the respondents appear in the database using a user code.

# Results

## General characteristics of the study sample

Table 1 shows the general characteristics of the population. A total of 12 026 people completed the survey. We excluded 669 participants who identified themselves as heterosexuals, 38 because were older than 60 years old, and 1559 because of missing values on the variables of interest. The final sample size involved 9760 participants. The median age of the sample was 25 IQR [21,30] years and 53.2% were male. Half of participants reported to have some level of higher education (47.9%) and a third reported not having health insurance (29.3%). Regarding sexual orientation, people self-identifying as gay was predominant (43.7%), followed by people self-identifying as bisexual (25.7%). 6.7% reported having done sex work some time. In the last 12 months, 12.9% and 10.6% suffered from a chronic and/or an infectious disease, respectively.

## Mental health problems and discrimination experiences

Discrimination/violence was reported by 70.3% of individuals and mental health problems in the last 12 months were reported in 23.8% of respondents. Stratifying by sexual orientation, the prevalence of discrimination was higher in participants who identified themselves as gay (74.6%), followed by pansexual, asexual and other (73.9%), lesbian (70.6%), and bisexual (61.6%). Regarding mental health problems, the prevalence was higher in who identified themselves as pansexual, asexual, and other (39.8%), followed by bisexual (29.6%), lesbian (19.8%), and gay (19.6%).

## Prevalence of mental health problems in the last 12 months by characteristics of the study population

Table 2 presents the bivariate analysis that relates the general characteristics of the population to mental health. A statistically significant difference (p<0.05) was observed with all variables, except with race (p = 0.096). The prevalence of mental health problems was higher in women compared with men (27.2% vs 20.8%, p<0.001). Concerning sex work, 28.4% of individuals

**Table 1. General characteristics, mental health problems and discrimination experiences among Peruvian sexual minorities aged 18 to 60 years old (n = 9760).**

| Characteristics | N (%) |
|---|---|
| Age (median [IQR]) | 25 [21,30] |
| Sex | |
| Male | 5196 (53.2) |
| Female | 4564 (46.8) |
| Education level | |
| No university | 5081 (52.1) |
| University | 4679 (47.9) |
| Health insurance | |
| None | 2857 (29.3) |
| SIS | 1327 (13.6) |
| Other health insurances | 5576 (57.1) |
| Sexual orientation | |
| Gay | 4268 (43.7) |
| Lesbian | 2215 (22.7) |
| Bisexual | 2506 (25.7) |
| Pansexual/Asexual/Other | 771 (7.9) |
| Race | |
| Indian native | 515 (5.3) |
| Black | 470 (4.8) |
| White | 1729 (17.7) |
| Mestizo | 6804 (69.7) |
| Other | 242 (2.5) |
| Disability | |
| No | 9468 (97.0) |
| Yes | 292 (3.0) |
| Sex work at least once | |
| No | 9109 (93.3) |
| Yes | 651 (6.7) |
| Chronic diseases in the last 12 months | |
| No | 8493 (87.0) |
| Yes | 1267 (13.0) |
| Infectious diseases in the last 12 months | |
| No | 8704 (89.2) |
| Yes | 1056 (10.8) |

IQR = Interquartile range.

SIS = Peruvian Comprehensive Health Insurance.

who engaged in such practices reported mental health problems, and in those who did not, the prevalence of problems was lower (23.5%). The prevalence of mental health problems in participants who reported discrimination and/or violence experiences was almost double compared with those who did not (27.2% vs 15.8%, p<0.001).

## Association between mental health problems in the last 12 months and discrimination experiences

The prevalence of mental health problems in the last 12 months was 72% higher in the group who experienced discrimination compared with the group that did not experience

**Table 2. Mental health problems in the last 12 months by characteristics of the Peruvian sexual minority aged 18 to 60 years old (n = 9760).**

| Variables | Mental health problems in the last 12 months | | P* |
|---|---|---|---|
| | **No** | **Yes** | |
| | **N = 7438** | **N = 2322** | |
| Discrimination and/or violence experiences | | | <0.001 |
| No | 2438 (84.2%) | 458 (15.8%) | |
| Yes | 5000 (72.8%) | 1864 (27.2%) | |
| Age (median [IQR]) | 25 [22,30] | 23.0 [20,28] | <0.001 |
| Sex | | | <0.001 |
| Male | 4115 (79.2%) | 1081 (20.8%) | |
| Female | 3323 (72.8%) | 1241 (27.2%) | |
| Education level | | | <0.001 |
| No one/Primary/Secondary | 3667 (72.2%) | 1414 (27.8%) | |
| Superior | 3771 (80.6%) | 908 (19.4%) | |
| Sexual orientation | | | <0.001 |
| Gay | 3433 (80.4%) | 835 (19.6%) | |
| Lesbian | 1776 (80.2%) | 439 (19.8%) | |
| Bisexual | 1765 (70.4%) | 741 (29.6%) | |
| Pansexual/Asexual/Other | 464 (60.2%) | 307 (39.8%) | |
| Race | | | 0.096 |
| Indian native | 379 (73.6%) | 136 (26.4%) | |
| Black | 353 (75.1%) | 117 (24.9%) | |
| White | 1358 (78.5%) | 371 (21.5%) | |
| Mestizo | 5166 (75.9%) | 1638 (24.1%) | |
| Other | 182 (75.2%) | 60 (24.8%) | |
| Disability | | | <0.001 |
| No | 7304 (77.1%) | 2164 (22.9%) | |
| Yes | 134 (45.9%) | 158 (54.1%) | |
| Sex work at least once | | | 0.005 |
| No | 6972 (76.5%) | 2137 (23.5%) | |
| Yes | 466 (71.6%) | 185 (28.4%) | |
| Health insurance | | | <0.001 |
| None | 2039 (71.4%) | 818 (28.6%) | |
| SIS | 995 (75.0%) | 332 (25.0%) | |
| Other health insurances | 4404 (79.0%) | 1172 (21.0%) | |
| Chronic diseases in the last 12 months | | | <0.001 |
| No | 6540 (77.0%) | 1953 (23.0%) | |
| Yes | 898 (70.9%) | 369 (29.1%) | |
| Infectious diseases in the last 12 months | | | 0.012 |
| No | 6600 (75.8%) | 2104 (24.2%) | |
| Yes | 838 (79.4%) | 218 (20.6%) | |

IQR = Interquartile range.

SIS = Peruvian Comprehensive Health Insurance.

*p values were calculated by Pearson's Chi-squared test for categorical variables and by Wilcoxon rank sum test with continuity correction for the age.

discrimination (APR = 1.72, 95% CI 1.57–1.88). Stratifying by sexual orientation, the association was stronger in the people self-identifying as gay and lesbian and a little weaker in people self-identifying as bisexual and pansexual/asexual/other. Those who identify themselves as

**Table 3. Association between discrimination experiences and mental health problems in the last 12 months in the Peruvian sexual minorities.**

| | Prevalence of mental health problems | Unadjusted [a] | | | Adjusted [a, b] | | |
|---|---|---|---|---|---|---|---|
| | N (%) | PR | 95% CI | p | PR | 95% CI | p |
| *All sample (n = 9760)* | | | | | | | |
| Without discrimination and/or violence experiences | 473 (15.4%) | ref. | | | ref. | | |
| With discrimination and/or violence experiences | 1929 (27.2%) | 1.72 | 1.57–1.88 | <0.001 | 1.72 | 1.57–1.88 | <0.001 |
| *Gay (n = 4268)* | | | | | | | |
| Without discrimination and/or violence experiences | 125 (11.6%) | ref. | | | ref. | | |
| With discrimination and/or violence experiences | 710 (22.3%) | 1.93 | 1.62 – 2.30 | <0.001 | 1.8 | 1.51 – 2.15 | <0.001 |
| *Lesbian (n = 2215)* | | | | | | | |
| Without discrimination and/or violence experiences | 71 (10.9%) | ref. | | | ref. | | |
| With discrimination and/or violence experiences | 368 (23.5%) | 2.16 | 1.70 – 2.73 | <0.001 | 2.08 | 1.65 – 2.64 | <0.001 |
| *Bisexual (n = 2506)* | | | | | | | |
| Without discrimination and/or violence experiences | 202 (21.0%) | ref. | | | ref. | | |
| With discrimination and/or violence experiences | 539 (34.9%) | 1.66 | 1.44 – 1.91 | <0.001 | 1.63 | 1.42 – 1.88 | <0.001 |
| *Pansexual/Asexual/Other (n = 771)* | | | | | | | |
| Without discrimination and/or violence experiences | 60 (29.9%) | ref. | | | ref. | | |
| With discrimination and/or violence experiences | 247 (43.3%) | 1.45 | 1.15 – 1.83 | 0.002 | 1.39 | 1.11– 1.74 | 0.004 |

[a] PR calculated using Poisson regression with a robust error variance.

[b] Prevalence ratio was adjusted for age, education level, health insurance, sex, race, disability, sex work, chronic diseases, infectious diseases.

95% CI = 95% confidence interval.

lesbian and experienced discrimination had more than twice the prevalence of mental health problems in the last 12 months (APR = 2.08, 95% CI 1.65–2.64) compared with the overall average. Table 3 presents all the results of the multivariable analysis.

In Table 4, we tabulated the PR of the model with interaction. In this model, the effect of discrimination on mental health problems increased (APR, 2.87; 95% CI, 1.70–4.86). People self-identified as gay, lesbian, bisexual, and pansexual who experienced discrimination had a 3.01, 2.38, 3.50, and 4.23 times higher prevalence of having mental health problems than heterosexuals who did not experience discrimination (Table 4).

## Discussion

This is the first study that assessed the association between discrimination and mental health in a sexual minority Peruvian population. Previous studies have shown that discrimination is associated with internalization disorders, i.e. mental health disorders, and externalization disorders, such as substance use disorders, in sexual minorities [26]. In the context of our study, we only addressed the mental health problems reported by the population studied, which was defined as having presented problems of depression or anxiety in the last 12 months.

Interestingly, the prevalence of discrimination was reported by 70% of the individuals. In England, Jackson et al., found the rate of perceived discrimination in adults self-identifying as gay, lesbian, and bisexual was 47.4%, even this reported that 23.7% received poor treatment by doctors or hospitals [27]. The difference between the prevalence of discrimination may be due to in developing countries the probability of acceptance of this sexual minority is lower [28]. Particularly, this alarming rate could be explained analyzing the Peruvian society context where acts of discrimination, in general, are quite common. In Peru, since 2017, as promulgated in the Legislative Decree 1323, acts of discrimination are forbidden, as well as incitement to discrimination based on gender identity and sexual orientation. Nevertheless, the

**Table 4. Association between discrimination on mental health problems in the last 12 months adding the interaction of sexual orientation.**

|  | PR [a] | 95% CI | p |
|---|---|---|---|
| *Discrimination and/or violence experiences* |  |  |  |
| No | Ref. |  |  |
| Yes | 2.87 | 1.70–4.86 | <0.001 |
| *Sexual orientation* |  |  |  |
| Heterosexual | Ref. |  |  |
| Gay | 1.64 | 0.98–2.74 | 0.059 |
| Lesbian | 1.15 | 0.68–1.96 | 0.604 |
| Bisexual | 2.14 | 1.30–3.51 | 0.003 |
| Pansexual/Asexual/Other | 3.07 | 1.82–5.18 | <0.001 |
| *With interaction* |  |  |  |
| Discrimination experience * Heterosexual | Ref. |  |  |
| Discrimination experience * Gay | 0.64 | 0.37–1.11 | 0.109 |
| Discrimination experience * Lesbian | 0.72 | 0.40–1.23 | 0.261 |
| Discrimination experience * Bisexual | 0.57 | 0.33–0.99 | 0.044 |
| Discrimination experience * Pansexual/Asexual/Other | 0.48 | 0.29–0.84 | 0.011 |

[a] PR calculated using Poisson regression with a robust error variance. The model included the interaction between sexual orientation and discrimination and adjusted for age, education level, health insurance, sex, race, disability, sex work, chronic diseases, infectious diseases.

95% CI = 95% confidence interval.

prevalence of discrimination remains high. This is a permanent challenge for organizations such as the Homosexual Movement of Lima and More Equality Peru, that have sought not only to defend LGTBI victims of discrimination, even to raise awareness in Peruvian society about the damage these acts of discrimination produce in all aspects of their lives. In Peru, mainly the media contribute to reducing discrimination against the LGTBI community, which have received a lot of rejection, however, nowadays they have achieved higher acceptation and awareness about the consequences of discrimination and also, in several open forums, respect for the LGTBI community is highlighted

Previous studies have shown that the LGBT community, compared to heterosexual population, have lower scores on several mental health indicators such as the remarkable higher need and use of mental health services, higher levels of smoking, as well as psychiatric diagnoses such as major depression, generalized anxiety disorders, and substance abuse [29–31]. In the present study, 23.8% of the analyzed population had mental health problems. In Ethiopia,in a population identified as gay, lesbian, and bisexual, 10.7% had depression, 14% stress and 20.8% anxiety or panic attacks [32].

Logie et al. evaluated the relationship between sexual stigma and depression in the LGBT population, and they found that 60.1% of respondents reported symptoms of depression. They also found low social support, low self-esteem, and economic insecurity mediated the relationship between sexual stigma and depressive symptoms [33].

Concerning anxiety, previous literature suggests that chronic experiences of discrimination based on sexual orientation may negatively alter the ability to regulate emotional responses, which may ultimately contribute to psychological symptoms and disorders [34]. In addition, tolerance to distress has been studied as a key vulnerability factor associated with high anxiety symptoms [35].

In the multivariable analysis, the discriminated sexual minority population had a 72% higher prevalence of mental health problems compared to the group that did not report discrimination. This association could be mediated by higher levels of victimization, depressive symptoms, and suicidal tendencies [36]. Other important factors are the self-concealment of sexual identity, the expectations of rejection, the lack of social support and the internalized homophobia [12,37]. Additionally, other factors such as discrimination type and frequency could be implicated in the association with mental health. A previous study in the LGBT population found that perceived discrimination in everyday life was associated with higher odds of depressive symptoms (OR = 2.30, 95% CI 1.02 to 5.21), loneliness (OR = 3.37, 95% CI 1.60 to 7.10), and lower quality of life (B = -3.31, 95% CI -5.49 to -1.12) [27]. Other study estimated that the odds of suffering any mental health disorder among those who reported gender discrimination was 2.28 times (AOR = 2.28, 95% CI 1.09 to 4.78) that of those who reported no discrimination. They also found that those individuals who suffered discrimination due to sexual orientation, gender discrimination, and racial discrimination were 3.31 times more likely to present any mental health disorder compared to those who did not report discrimination (AOR = 3.31, 95% CI 1.45 to 6.74) [9]. These findings suggest that within the sexual minority community there are subpopulations at risk for other types of discrimination that may increase the likelihood of mental health disorders. However, in the present study, we did not evaluate the type of discrimination or the extent and frequency of discrimination.

Another interesting finding in the multivariable subgroup analysis was that people who identified as lesbians were the population most affected by discrimination (APR = 2.08, 95% CI 1.65–2.64), followed by people who identified as gay (APR = 1.80, 95% CI 1.51–2.15). Despite the different studies that have been carried out to understand how discrimination affects mental health in the LGBT population, most chose only to analyze the association between discrimination and mental health in the entire LGBT sample [9,30]. Few studies carry out analyses among subgroups, i.e., among the population self-identifying as lesbian, gay, bisexual, and pansexual. This makes it difficult to compare the results found with previous literature. More studies that analyze how discrimination affects each of the aforementioned subgroups and the difference between these are needed because the social interactions of each subgroup are different, i.e. transgenders are not too accepted as gay populations among Peruvian LGTB community and the rest of society. In addition, knowledge of the characteristics and implications of discrimination in each subgroup would lead to proposal future strategies for an adequate and timely approach to the mental problems associated with discrimination in the LGBT community.

All the studies mentioned agree that discrimination in the sexual minority population leads to mental health problems [9,26–37]. In the present study, the effect of discrimination on mental health found in the general sample was not as high as in the studies cited. These differences may be since most previous studies have used odds ratios to measure the effect. However, when the prevalence of the outcome of interest in the population is greater than 10%, this statistical test tends to overestimate the effect. Another possible explanation is that the effect found is different from the real one because of other unmeasured variables such as internalized homonegativity, sensitivity to rejection, social support, social network, belonging to an LGB community, perceived stress, and child gender dissatisfaction [38].

In the present study, we evidence the association between discrimination and self-reported mental health problems. New strategies focused on reducing discrimination in these vulnerable populations should be implemented, since the current ones are apparently not effective. Especially in those subpopulations with a greater association with mental health problems should be protected. We recommend that future studies explore this association evaluating the type, frequency, and extent of the discrimination. Also, mental health problems should be

evaluated with validated tools for depression (PHQ-9) and anxiety (GAD-7). Other outcomes such as suicidal ideation could be addressed. Ideally, prospective studies should be performed.

## Strengths and limitations

Some limitations must be highlighted. First, this is a secondary analysis, which not only focuses on mental health and discrimination topics; thus, some variables have not been considered. Second, we could not assess causality between the evaluated variables due to the cross-sectional nature of our study. Third, a non-probabilistic sampling was made due to no prior information being available on the size of this population to select a sample. Fourth, the survey had no details regarding the type, frequency, or extent of the discrimination. Fifth, the mental health variable was by self-report and the survey did not include a validated instrument to detect mental health symptoms. Nevertheless, despite these limitations, this study analyzes the first national survey of the Peruvian sexual minority population and has a large sample. Thus, our findings could serve as the basis for developing and strengthening public health policies to decrease discrimination in sexual minority populations in Peru.

## Conclusion

This study found a high prevalence of discrimination and mental health problems in the last 12 months in the sexual minority population. Additionally, an association between both conditions was determined. The problem of discrimination in this population was very common, demonstrating the need for educational and socio-cultural interventions that can reduce this alarming figure. Furthermore, this population presented a high prevalence of mental health problems so it is important to increase efforts for the diagnosis and necessary treatment of these conditions. In addition, it is recommended that new prospective studies be carried out as far as possible to determine other risk factors, as well as to evaluate other mental health problems.

## Acknowledgments

Special thanks to Michael White, Universidad Peruana Unión, who provided much support by reviewing the draft of this article.

## Author Contributions

**Conceptualization:** David R. Soriano-Moreno, David Saldaña-Cabanillas, Luigy Vasquez-Yeng, Javier Antonio Valencia-Huamani, Jorge Luis Alave-Rosas, Anderson N. Soriano.

**Formal analysis:** Anderson N. Soriano.

**Methodology:** David R. Soriano-Moreno, David Saldaña-Cabanillas, Luigy Vasquez-Yeng, Javier Antonio Valencia-Huamani, Jorge Luis Alave-Rosas, Anderson N. Soriano.

**Writing – original draft:** David R. Soriano-Moreno, David Saldaña-Cabanillas, Luigy Vasquez-Yeng, Javier Antonio Valencia-Huamani, Jorge Luis Alave-Rosas, Anderson N. Soriano.

**Writing – review & editing:** David R. Soriano-Moreno, David Saldaña-Cabanillas, Luigy Vasquez-Yeng, Javier Antonio Valencia-Huamani, Jorge Luis Alave-Rosas, Anderson N. Soriano.

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
