## [Decision Letter · Decision Letter 0]

14 Dec 2021

PONE-D-21-30055Discrimination and Mental Health in the minority sexual population: Cross-sectional analysis of the First Peruvian Virtual SurveyPLOS ONE

Dear Dr. Soriano,

Thank you for submitting your manuscript to PLOS ONE. After careful consideration, we feel that it has merit but does not fully meet PLOS ONE’s publication criteria as it currently stands. Therefore, we invite you to submit a revised version of the manuscript that addresses the points raised during the review process.

We look forward to receiving your revised manuscript.

Kind regards,

Bidhubhusan Mahapatra, Ph.D.

Academic Editor

PLOS ONE

Journal Requirements:

Additional Editor Comments (if provided):

This is an interesting paper and provide important insights into the mental health issues among sexual minority population. I have couple of observations:

1. Provide more details on the data collection process, its management and quality assurance process. Clarify, if any incentive was provided to participants.

2. I feel the Table 2 is not adding any value when I look at Table 3. You should include a table that provides prevalence of mental health problems by experience of discrimination for each category of sexual identity that would enable making sense of odds ratios presented in Table 4.

3. I think discussion is missing the critical interpretation on what does the prevalence of mental health and discrimination mean for Peru and programs working on sexual minorities, how the high prevalence of mental health higher for some groups even if the prevalence of discrimination is lower? There are multiple things to discuss.

Reviewers' comments:

Reviewer's Responses to Questions

**Comments to the Author**

1. Is the manuscript technically sound, and do the data support the conclusions?

Reviewer #1: Partly

Reviewer #2: Partly

2. Has the statistical analysis been performed appropriately and rigorously? 

Reviewer #1: I Don't Know

Reviewer #2: No

3. Have the authors made all data underlying the findings in their manuscript fully available?

Reviewer #1: No

Reviewer #2: Yes

4. Is the manuscript presented in an intelligible fashion and written in standard English?

Reviewer #1: No

Reviewer #2: No

5. Review Comments to the Author

Reviewer #1: Additional comments: It would be helpful to know if any of the research team/authors on the paper, are part of the sexual or gender minority populations under study (in alignment with the notion "nothing for us without us"). Some of the language, as written, comes off as "othering", so I suggest a modest revision when describing populations of transgender experience, and less use of the words they/them in reference (although this is could be an interpretation). Explicitly state in the abstract and the intro or methods that this is a secondary analysis for further clarity on the methodology and type of study undertaken. This is a very important topic and I am grateful this research was undertaken - it is potentially lifesaving. Clarity is needed on the definition of "mental health problems" as only including experience of depression and/or anxiety - did this mean by a clinical definition of depression and/or anxiety inclusive of severe symptoms such as panic attacks, or just generally that the person reported feeling depressed or anxious in the past year (as these are drastically different since all humans experience depression and/or anxiety and are not "mental health problems" as a stand alone, but whether or not symptoms are debilitating and chronic - this needs to be clarified from what the original survey intended). Please provide further clarification on justifying the two different time frames used in the original survey, and how this was accounted for in your secondary analysis regarding association: having "ever" experienced discrimination and experienced mental health problems "in the past 12 months." For the discrimination variable(s), is there a scale or additional questions to specify the type/experience/extent/frequency of discrimination experienced? This seems important to understand better if the suggestion is a direct association between discrimination (as the exposure) and mental health problems, and is it possible to make this association in a non-randomized sample - or do the interpretation of findings suggest that the "exposure" of discrimination doesn't matter for it to impact anxiety and depression specifically? This should be clarified. Consider adding a sample of the original survey questions that were asked for discrimination and mental health to make these points more clear. The majority of the Discussion is focused on previous studies in different countries, and little on the findings of this secondary analysis. The background literature is useful for context, but not for comparison of findings. Consider revising the Discussion to go into more detail about the findings from the secondary analysis. Limitation of not being able to assess causality was listed, but the findings seem to imply that ever experiencing discrimination is associated with recent anxiety and/or depression; suggest elaborating on what this association may imply and to make suggestions on further research needed to explore this further (since not possible to know causality at this stage/with these methods) - elaborate on recommendations as written, they are too vague especially when referring to the need for "new prospective studies" [276-277]. This is a very important study with important findings on a population in need of support of services - recommendation is to revise and resubmit.

Reviewer #2: This paper examines the relationship between discrimination and mental health in a Peruvian sexual minority population. This is an important topic that merits additional research. Below I provide several comments that can help strengthen the analysis and writing of the paper.

Introduction:

- The authors provided a lot of context for sexual minority in the U.S. but very little for those in Peru. More of the social and historical context for Peru (the study setting) is needed.

- I strongly advise the authors to consider the use of a theory or conceptual framework for their study question.

Methods:

- More information is needed about how this sample of 12000+ sexual minority individuals was recruited.

- Please justify the use of the non-validated question to measure mental health. There are several options that could have been used (e.g., PHQ-9). The use of unvalidated instrument is a significant limitation of this paper.

- Please justify why the question about experiences of discrimination only asked about general discrimination and did not specify discrimination based on sexual orientation.

- Please harmonize and be consistent about how you present the questions and the answer options. Currently some are presented in parentheses, some in a different sentence, etc.

- Please correct “multivariate” to “multivariable” throughout the paper.

Results:

- Please be mindful of some language (e.g., you should say “people self-identifying as gay” instead of “the gay population”). Please be consistent with capitalizing or not capitalizing sexual orientation (e.g., Gay, Lesbian)

- The table 4’s header does not make it clear that the outcome is mental health. I suggest a revision.

- A major issue is the subgroup analysis. You are analyzing data separately for subsamples and then comparing effect sizes of separate subsamples. This is not an appropriate practice – I suggest instead the use of interaction terms.

Discussion:

- Similar to the introduction, more discussion of the implications of your findings in the Peruvian context is needed.

6. PLOS authors have the option to publish the peer review history of their article (what does this mean?). If published, this will include your full peer review and any attached files.

Reviewer #1: No

Reviewer #2: No

---

## [Author Response · Author response to Decision Letter 0]

16 Mar 2022

Response to Reviewers

Journal Requirements:

Reply: Thanks for the observation, we modified the style according to the journal.

2. We note that the grant information you provided in the ‘Funding Information’ and ‘Financial Disclosure’ sections do not match. When you resubmit, please ensure that you provide the correct grant numbers for the awards you received for your study in the ‘Funding Information’ section

Reply: Thank you. We have added the number of the grant that supports the corresponding author. 

EDITOR:

1. Comment: Provide more details on the data collection process, its management and quality assurance process. Clarify, if any incentive was provided to participants.

Reply:

Thank you for the comment. We have detailed the data collection process, its management and quality assurance process in the methods section using the details that the Peruvian National Institute of Statistics brings on their website.

2. Comment: I feel the Table 2 is not adding any value when I look at Table 3. You should include a table that provides prevalence of mental health problems by the experience of discrimination for each category of sexual identity that would enable making sense of odds ratios presented in Table 4.

Reply: Thank you, we have added the prevalence of mental health problems by the experience of discrimination for each subgroup in a new table 3 and we deleted the past table 2.

3. Comment: I think discussion is missing the critical interpretation on what does the prevalence of mental health and discrimination mean for Peru and programs working on sexual minorities, how the high prevalence of mental health higher for some groups even if the prevalence of discrimination is lower? There are multiple things to discuss.

Reply: We thank the comment, to solve this lack of context, we added the following in the discussion: “Particularly, this alarming figure could be explained observing the context that surrounds Peruvian society where acts of discrimination, in general, are quite common. In Peru, since 2017, as promulgated in Legislative Decree1323, acts of discrimination are prohibited as well as incitement to discrimination based on gender identity and sexual orientation. Nevertheless, even with this measure, we observe that the prevalence of discrimination remains high. This is a constant challenge for associations such as the Homosexual Movement of Lima and More Equality Peru that have sought not only to defend LGTBI victims of discrimination but also to raise awareness in Peruvian society of the damages that these acts of discrimination generate in health and integral life these acts of discrimination. Important support in Peru to reduce discrimination against the LGTBI community is the media, which began receiving a lot of rejection, however, today they have achieved a better understanding of the consequences of discrimination and in the different open forums, comments improve and respect for the LGTBI community is encouraged.”

REVIEWER #1: 

4. Comment: It would be helpful to know if any of the research team/authors on the paper, are part of the sexual or gender minority populations under study (in alignment with the notion "nothing for us without us"). 

Reply: In the research team, none of the authors is part of the sexual minority population.

5. Comment: Some of the language, as written, comes off as "othering", so I suggest a modest revision when describing populations of transgender experience, and less use of the words they/them in reference (although this is could be an interpretation). 

Reply: Thanks for the suggestion. We changed the sentence in the introduction to avoid using they/them: “This is a very difficult situation because this population may also suffer discrimination when reporting. [10]”

6. Comment: Explicitly state in the abstract and the intro or methods that this is a secondary analysis for further clarity on the methodology and type of study undertaken. This is a very important topic and I am grateful this research was undertaken - it is potentially lifesaving. 

Reply: We appreciate the comments, and we added this information in the abstract and in the methods. Abstract: “We conducted a cross-sectional analysis of a secondary database corresponding to the first LGBTI survey in Peru in 2017.” Methods: “This study is a cross-sectional analysis of a secondary database from the First Virtual Survey for LGTBI people in Peru carried out by the Peruvian National Institute of Statistics and Informatics (INEI) in 2017.”

7. Comment: Clarity is needed on the definition of "mental health problems" as only including experience of depression and/or anxiety - did this mean by a clinical definition of depression and/or anxiety inclusive of severe symptoms such as panic attacks, or just generally that the person reported feeling depressed or anxious in the past year (as these are drastically different since all humans experience depression and/or anxiety and are not "mental health problems" as a stand alone, but whether or not symptoms are debilitating and chronic - this needs to be clarified from what the original survey intended). 

Reply: Thank you, we clarified this in the variables section in methods. We added, “This question involves any degree of symptomatology from a belief of anxiety or depression to a clinical diagnosis.” Regarding the terminology, we use the “mental health problem” term throughout the manuscript because it is the one mentioned by the question with which the outcome was evaluated.

8. Comment: Please provide further clarification on justifying the two different time frames used in the original survey, and how this was accounted for in your secondary analysis regarding association: having "ever" experienced discrimination and experienced mental health problems "in the past 12 months." For the discrimination variable(s), is there a scale or additional questions to specify the type/experience/extent/frequency of discrimination experienced? This seems important to understand better if the suggestion is a direct association between discrimination (as the exposure) and mental health problems, and is it possible to make this association in a non-randomized sample - or do the interpretation of findings suggest that the "exposure" of discrimination doesn't matter for it to impact anxiety and depression specifically? This should be clarified. Consider adding a sample of the original survey questions that were asked for discrimination and mental health to make these points more clear. 

Reply: We added in the methodology that: “Given that there was no question used to measure the type, the frequency and extent of the discrimination, the understanding is that respondents would report any life experience.”

9. Comment: The majority of the Discussion is focused on previous studies in different countries, and little on the findings of this secondary analysis. The background literature is useful for context, but not for comparison of findings. Consider revising the Discussion to go into more detail about the findings from the secondary analysis. 

Reply: In the discussion, we added more context about the Peruvian setting and the implications of the results.

10. Comment: Limitation of not being able to assess causality was listed, but the findings seem to imply that ever experiencing discrimination is associated with recent anxiety and/or depression; suggest elaborating on what this association may imply and to make suggestions on further research needed to explore this further (since not possible to know causality at this stage/with these methods) - elaborate on recommendations as written, they are too vague especially when referring to the need for "new prospective studies" [276-277]. This is a very important study with important findings on a population in need of support of services - recommendation is to revise and resubmit.

Reply: We appreciate the suggestion, we added recommendations for future research at the end of the discussion.

REVIEWER #2: 

11. Comment: Introduction: The authors provided a lot of context for sexual minority in the U.S. but very little for those in Peru. More of the social and historical context for Peru (the study setting) is needed.

Reply: Thanks for the observation, we added the following sentences in the introduction to fill this gap. “In Peru, since January 2017 Legislative Decree 1323 entered into force, including as categories protected against discrimination, gender identity, and sexual orientation, being also considered as aggravating in crimes against LGBTI persons. (("Peruvian Penal Code 2021 updated"))” and “In Peru, a country with conservative traits, unions between people of the same sex has not yet been officially recognized. Most of the population rejects the implementation of equal marriage. With a government that prohibits discrimination and incitement to discrimination based on sexual orientation and gender identity, but that still maintains Article 333 subsection 9 of the current Civil Code, where homosexuality is considered as a ground for divorce. It fails to curb the high rates of discrimination suffered by the LGBTI community.”.

12. Comment: Introduction: I strongly advise the authors to consider the use of a theory or conceptual framework for their study question.

Reply: We appreciate the suggestion. We used the Meyer conceptual framework (Minority stress processes in lesbian, gay, and bisexual populations).

13. Comment: Methods: More information is needed about how this sample of 12000+ sexual minority individuals was recruited.

Reply: Thank you. We have added in the method section the details about data collection.

14. Comment: Methods: Please justify the use of the non-validated question to measure mental health. There are several options that could have been used (e.g., PHQ-9). The use of unvalidated instrument is a significant limitation of this paper.

Reply: Thank you. Yes, is an important limitation. We have clarified this in the limitations section. However, there is no other available database with data of this nature.

15. Comment: Methods: Please justify why the question about experiences of discrimination only asked about general discrimination and did not specify discrimination based on sexual orientation.

Reply: Thank you. The survey and dataset do not include details about the cause of discrimination. That is why we could not evaluate that aspect. We added this concern to the limitations section.

16. Comment: Methods: Please harmonize and be consistent about how you present the questions and the answer options. Currently some are presented in parentheses, some in a different sentence, etc.

Reply: We modified the sentences to use only parentheses. 

17. Comment: Methods: Please correct “multivariate” to “multivariable” throughout the paper. 

Reply: We changed the term to multivariable. 

18. Comment: Results: Please be mindful of some language (e.g., you should say “people self-identifying as gay” instead of “the gay population”). Please be consistent with capitalizing or not capitalizing sexual orientation (e.g., Gay, Lesbian)

Reply: We agree with the comment, we homogenize the terms throughout the manuscript.

19. Comment: Results: The table 4’s header does not make it clear that the outcome is mental health. I suggest a revision.

Reply: Thank you. We have clarified the outcome on the new table 3.

20. Comment: Results: A major issue is the subgroup analysis. You are analyzing data separately for subsamples and then comparing the effect sizes of separate subsamples. This is not an appropriate practice – I suggest instead the use of interaction terms.

Reply: Thank you for the suggestion. We have complemented the analysis with a regression model with interaction to evaluate the effect of discrimination and sexual identity on mental health problems (table 4). 

Comment: Discussion: Similar to the introduction, more discussion of the implications of your findings in the Peruvian context is needed.

Reply: Thanks for the comment. We added more details about the Peruvian context and the implications of the results.

---

## [Editor Report · Decision Letter 1]

9 May 2022

Discrimination and Mental Health in the minority sexual population: Cross-sectional analysis of the First Peruvian Virtual Survey

PONE-D-21-30055R1

Dear Dr. Soriano,

We’re pleased to inform you that your manuscript has been judged scientifically suitable for publication and will be formally accepted for publication once it meets all outstanding technical requirements.

Kind regards,

Bidhubhusan Mahapatra, Ph.D.

Academic Editor

PLOS ONE
---

## [Editor Report · Acceptance letter]

26 May 2022

PONE-D-21-30055R1 

Discrimination and Mental Health in the minority sexual population: Cross-sectional analysis of the First Peruvian Virtual Survey 

Dear Dr. Soriano:

I'm pleased to inform you that your manuscript has been deemed suitable for publication in PLOS ONE. Congratulations! Your manuscript is now with our production department. 

Kind regards, 

on behalf of

Dr. Bidhubhusan Mahapatra 

Academic Editor

PLOS ONE